# Biocide Activity of Green Quercetin-Mediated Synthesized Silver Nanoparticles

**DOI:** 10.3390/nano10050909

**Published:** 2020-05-08

**Authors:** Federico Tasca, Riccarda Antiochia

**Affiliations:** 1Departamento de Química de los Materiales, Facultad de Química y Biología, Universidad de Santiago de Chile, Av. Libertador Bernardo O´Higgins 3363, Santiago 9170022, Chile; federico.tasca@usach.cl; 2Department of Chemistry and Drug Technologies, Sapienza University of Rome, P.zale Aldo Moro 5, 00185 Rome, Italy

**Keywords:** green synthesis, silver nanoparticles, quercetin, biocide activity

## Abstract

The development of new nanomaterials is gaining increasing attention due to their extensive applications in fields ranging from medicine to food and cultural heritage. Green nanoparticles provide advantages compared to conventional nanoparticles as their synthesis is environmentally-friendly and does not require the use of high temperatures, pressure, or toxic chemicals. In this paper, green silver nanoparticles (AgNPs) have been synthesized according to a new method using quercetin as a reducing agent at room temperature. The synthesized AgNPs were characterized using UV-Vis spectroscopy, transmission electron microscopy (TEM), energy dispersive spectroscopy (EDS), and dynamic light scattering (DLS) techniques and successively tested for biocide activity by studying their effects in the inhibition of bacterial growth. The results demonstrated that the smaller the AgNPs size, the greater their biocide activity. In particular, AgNPs with a diameter of 8 nm showed a minimum inhibitory concentration (MIC) value of 1.0 μg/mL against *Streptococcus* sp., *Escherichia coli* and *Candida* sp. microorganisms, while AgNPs with a larger diameter of about 20 nm were able to inhibit microbial of all selected pathogens at a higher MIC value of 2.5 μg/mL.

## 1. Introduction

Nanotechnology, a new revolutionary area of modern research, is helping to improve and, in some cases, revolutionize many industry sectors, such as medicine, information technology, energy, food safety, sensing devices, and environmental science. In recent years, possible applications of metal nanoparticles and nanomaterials have been studied in the field of conservation of cultural heritage. Highly specialized methods are necessary for both diagnostics and treatment of monuments and/or artefacts [1]. Thanks to their excellent properties, such as small size, high penetrability, thermal stability, biological and chemical inertness, non-toxic, and magnetic properties, compared to traditional materials, nanomaterials can offer viable solutions to several problems regarding the degradation of cultural patrimony [2]. Nanomaterials are defined as nanosized metals with dimensions ranging from 1 to 100 nm. The two main areas of utilization of nanomaterials and metal nanoparticles are: (i) as nano-consolidants, in the treatment of surfaces (protection, waterproofing, etc.) and as new products for consolidation of stone materials; and (ii) as biocides for cleaning surfaces affected by polluting substances and biological agents.

It is well known that monuments and cultural heritage structures are affected by degradation processes caused by temperature variations, moisture, and pollution. Microorganisms also play an important role in the so-called biodeterioration process. Biodeterioration can be defined as the set of damages caused by the action of biological agents, and it is estimated that it is responsible for about 30% of the total degradation process of stone buildings [3,4]. Microorganisms are responsible for two different alteration mechanisms: (a) a physical alteration process which results in the loss of cohesion, generally due to mode of adhesion of the microorganisms to the surface; (b) a chemical alteration process, where a chemical reaction is involved and causes the decomposition of the substrate due to the metabolisms of the microorganisms which consume and excrete different substances with specific chemical reactivity [2].

It is known that particular metals such as Cu and Zn show biocide properties against microorganisms such as cyanobacteria, fungi, algae, etc. [5]. Unfortunately, their use is limited because of their high solubility, with consequent leaching from the surfaces. Cu salts have also problems relating to their intense blue colour. The use of silver as a disinfectant agent is also known from ancient times [2]. In recent decades, its use has been progressively reduced because of the introduction of antibiotics. However, with the advent of nanotechnology, the use of silver on a nanometric scale, such as with silver nanoparticles, has reinserted its use in several fields. By combining the antiseptic properties of silver with the nano-properties of nanoparticles, the silver nanoparticles (AgNPs) are generating large interest for their successful applications [6,7,8].

The bactericidal mechanism of action of AgNPs has not been completely understood, but it is almost clear that its biocidal action is based on the capacity to damage both the bacteria cell wall and plasma membrane and also to inhibit DNA replication. It seems that Ag^+^ ions are responsible for the biocide effect as they are easily released from the surface of silver nanoparticles and the marked effect depends on the lower concentration of the Ag^+^ ions compared to normal silver salts. Moreover, the biocide effect seems to be strictly related to the nanoparticle size, shape and type of reducing agent used for their synthesis [9].

Another aspect to be considered in the conservation of cultural heritage is that the use of toxic chemicals is not recommended as they increase the risk of exposure of visitors. Therefore, the use of natural, eco-friendly products in the conservation processes is gaining increasing importance.

The integration of green chemistry with nanotechnology is an emergent topic of recent nanoscience research. The production of metal NPs through a green synthetic path shows great advantages thanks to its slower kinetics, easy manipulation, and the possibility to control crystal growth offering a sustainable and safe method [10,11]. Several plant extracts, fungi, and bacteria have been utilized for the green synthesis of MNPs showing numerous benefits for applications in different fields [12,13,14,15,16,17,18,19,20,21,22,23,24,25,26,27]. Actinobacteria [28] and lactid acid bacteria [29] were recently used as safe and inexpensive sources of bioactive silver nanoparticles. Among the biomolecules present in plant extracts it seems that proteins, polyphenols, sugars, and vitamins, such as L-ascorbic acid, caffeic acid, curcumin [30], and quercetin [31], showed the best results to reduce the metal ions to metal NPs. Quercetin (QUC, 3,3′,4′,5,7-pentahydroxyflavone) is a polyphenolic compound present in fruits, vegetables, and common beverages, such as red wine and green tea [32]. It shows also important antioxidant properties thanks to its ability in scavenging free radicals and in chelating metal ions [33].

The aims of the present paper are: (1) green synthesis of AgNPs by the reduction of silver ions using quercetin as reducing agent; (2) characterization of the AgNPs using UV-Vis spectrophotometry, transmission electron microscopy (TEM), dynamic light scattering (DLS) and energy dispersive spectroscopy (EDS); (3) study of the microorganisms inhibition capacity of the synthesized AgNPs to prevent and control the biodeterioration process in cultural heritage.

## 2. Experimental

### 2.1. Synthesis, Purification and Characterization of AgNPs

The synthesis of silver nanoparticles was carried out using quercetin (QUC) to reduce the metal precursor silver nitrate (AgNO_3_). Quercetin was solubilized in 1 M NaOH solution with a final concentration of 1 mM, then diluted with 50 mM phosphate buffer (PBS) at pH = 7 to a final concentration of 50 μM. AgNO_3_ was solubilized in 50 mM PBS buffer pH = 7. The QUC concentration of QUC was fixed at 50 µM while the concentration of AgNO_3_ was varied from 10 µM to 500 µM. The synthesis was realized at room temperature (20 °C) in 50 mM PBS buffer at pH = 7. The formation of a brown colour of the solution was an indicator of the formation of AgNPs.

The nanoparticles colloidal solution was centrifuged at 9000 rpm for 10 min, the supernatant was removed, and the precipitate was redispersed in Milli Q water and sonicated for 10 min for homogenization. The procedure was repeated twice for optimal purification.

QUC, AgNO_3_, sodium phosphate monobasic (NaH_2_PO_4_), and sodium phosphate dibasic (Na_2_HPO_4_) were obtained from Sigma Aldrich (Stockholm, Sweden). Milli Q water was obtained from Millipore (Bedford, MA, USA).

### 2.2. UV-Visible Spectral Analysis of AgNPs

Preliminary characterization of AgNPs was carried out using UV-Visible spectroscopy. The reduction of silver ions to form NPs was monitored by measuring the UV-Visible spectra of the solutions after diluting the sample with 50 mM PBS buffer at pH = 7. The kinetics of the reaction was monitored by recording UV-Visible spectra with time. The UV-Visible spectra were recorded on a Shimadzu UV-2401 PC spectrophotometer (Kyoto, Japan) with samples contained in a quartz cuvette operated at a resolution of 1 nm from 300 to 500 nm for quercetin and for AgNPs. The spectra were re-plotted using GraphPad Prism 6 (San Diego, CA, USA). The blank was performed using Milli-Q water.

### 2.3. Energy Dispersive Spectroscopy (EDS) Analysis of AgNPs

EDS experiments were carried out by using an energy dispersive detector on scanning electron microscopy (Zeiss EVO MA10 Oxford in., Bucks, UK). Two drops of the synthesized colloidal solution were deposited on a polymer thin copper grid; successively the solvent was evaporated under vacuum.

### 2.4. Transmission Electron Microscopy (TEM) Analysis of AgNPs

The characterization of the AgNPs in terms of shape and size was carried out by using Philips/FEI BioTwin CM120 TEM transmission electron microscopy (FEI, Hillsboro, OR, USA). Two drops of the synthesized colloidal solution were deposited on a polymer thin copper grid; successively the solvent was evaporated under vacuum.

### 2.5. Dynamic Light Scattering (DLS) Analysis of AgNPs

The particle size range of AgNPs, their polydispersity and the Z-potential were determined using a Zetasizer analyser (Zetasizer Nano ZS90, Malvern In. Ltd., Malvern, UK). The measurement of particle size was obtained by measuring the time-dependent fluctuation of the laser light scattering through the nanoparticles, while the measurement of the Z-potential was realized by measuring the direction and velocity of the nanoparticles under an electric field of known intensity.

### 2.6. Antimicrobial Activity of AgNPs

*Streptococcus* sp., *Escherichia coli* and *Candida* sp. were purchased from LGC Standards srl, Milan, Italy, to test the AgNPs antimicrobial activity. These microorganisms were chosen as typical examples, respectively, of positive bacteria, negative bacteria, and fungi, responsible for biodegradation and biodeterioration of monuments and cultural heritage sites. Sterile culture tubes with 5 mL of Luria-Bertani (LB) broth (casein peptone 10 g, sodium chloride 10 g, and yeast extract 5 g) were inoculated with 100 μL of selected bacteria and incubated at 37 °C for 48 h. Successively, 100 μL of AgNPs colloidal solutions obtained with 350 e 500 μM AgNO_3_ was added to the tubes and incubated at 37 °C for 24 h with shaking the solution at 150 rpm [7]. The microbial growth was followed by measuring the absorbance at 600 nm using a Shimadzu UV-2401 PC spectrophotometer (Kyoto, Japan).

## 3. Results and Discussion

### 3.1. Green Synthesis of AgNPs

AgNPs were synthesized using an environmentally-friendly biosynthetic method in order to reduce the use of organic solvents and to avoid the production of toxic waste. In particular, the synthesis was carried out by using quercetin in aqueous solutions as a reducing agent at room temperature, according to a procedure developed in our previous work [27]. The reaction of the reduction of Ag ions to form NPs is shown in Figure 1. Quercetin gives two electrons to Ag^+^ forming two Ag^0^ NPs, with the consequent oxidation of the catechol present on the B ring of the o-quinone group.

The main advantages of the quercetin synthesis method compared to other methods reported in the literature are: (i) no need of extreme experimental conditions because the experiment can be carried out at room temperature; (ii) high reaction yield; and (iii) short reaction time (20 min). Several “green” synthesis methods report on the synthesis using, e.g., starch and glucose as reducing agents at a high temperature for 20 h [21], or by using L-ascorbic acid at 80 °C [19,20], and other methods report on the need for microwave assistance [22,23]. Another important feature of the proposed method is that it is highly reproducible, especially compared to the methods based on the use of natural fruit extracts, where several variables are very difficult to control [24,25,26].

### 3.2. UV-Visible Spectral Characterization of AgNPs

Figure 2a shows the UV-VIS spectra of a colloidal solution of quercetin and AgNPs solution between 300 and 500 nm. It is possible to observe the two typical absorption bands for QUC at λ = 322 nm and λ = 360 nm (blue curve) and the AgNPs typical surface plasmon resonance (SPR) band at λ = 403 nm (red curve), as reported in the literature [20,27]. The UV-Vis spectrophotometry allowed us to monitor the synthesis of AgNPs by varying the concentration of the metal precursor. The AgNO_3_ concentration was varied in the range 50–500 μM in order to optimize the synthesis conditions. Figure 2b shows the UV-VIS spectra recorded at a fixed concentration of QUC (50 μM) in 50 mM PBS buffer at pH 7 and at increasing AgNO_3_ concentrations. It is possible to note that the absorbance progressively increases at increasing metal precursor concentrations from 50 up to 350 μM, while a decrease of absorbance was registered at higher concentrations. A concentration of 350 μM was therefore chosen as optimal metal precursor concentration for further measurements.

By carefully examining Figure 2b, we see that at increasing metal precursor concentrations, a slight shift to long wavelengths can be noted. In particular, at concentrations up to 350 mM, the spectra show the typical SPR band at about *λ* = 403 nm, while at higher AgNO_3_ concentrations a red shift in the absorption peaks with *λ* values up to 412 nm is registered. This result might be ascribed to the fact that the AgNO_3_ concentration can influence nanoparticles size, obtaining AgNPs with different sizes when different AgNO_3_ concentrations are employed in the synthesis. This is confirmed from data published in the literature, where it is reported that there is a direct relationship between the position of the absorption peak and particle size [8].

The reaction time was also studied by UV-VIS spectrophotometry by following the change in absorbance at λ = 403 nm for AgNPs versus time. The maximum absorbance value was obtained at 20 min with a concentration of 350 μM (data not shown).

Moreover, the concentration of AgNPs colloidal solution has been calculated by using an extinction molar coefficient of 4.18 × 10^9^ M^−1^·cm^−1^ at *λ*
_max_ = 405 nm [34] and resulted to be 0.71 and 0.42 μM for 350 and 500 μM metal precursor, respectively.

### 3.3. Energy Dispersive Spectroscopy (EDS) Analysis of AgNPs

The first step of the characterization of the AgNPs was the analysis using EDS spectroscopy in order to check the elemental composition of the colloidal suspensions of the synthesized AgNPs. It is possible to see in Figure 3 the typical pattern of AgNPs, as reported in the literature [27].

### 3.4. Transmission Electron Microscopy (TEM) Analysis of AgNPs

Successively, the characterization of shape, dimension, and size distributions of AgNPs was carried out by transmission electron microscopy (TEM). The measurements were performed with AgNPs solutions obtained with 350 and 500 μM metal precursor concentrations. The results are shown in Figure 4a,b. A quasi-homogeneous size distribution with spherical colloid particles and a crystalline structure with no remarkable aggregates or clusters is observed for both concentrations [21]. The average diameter of the nanoparticles resulted to be 8.4 ± 0.3 nm and 20.0 ± 0.4 nm for AgNPs synthesized with 350 and 500 μM metal precursor, respectively, as shown in Figure 4c,d, showing that the particles size is strongly influenced by metal precursor concentrations, as already hypothesized from the UV-VIS data (see Section 3.2). 

### 3.5. Particle Size and Zeta Potential Measurements of AgNPs

In order to confirm the results obtained with TEM measurements, dynamic light scattering (DLS) experiments were carried out, and the results are exhibited in Figure 5 for AgNPs samples obtained with a 350 μM AgNO_3_ concentration. These measurements allowed us to determine the hydrodynamic diameter, named Z-average [33], and the Z-potential, in order to evaluate the surface charge and the stability of the colloidal solution [34]. The hydrodynamic diameter of AgNPs resulted to be 10.0 ± 0.6 nm with a low polydispersion index (PDI) of 0.292 ± 0.049. It is interesting to note that this value is larger than the average diameter determined by TEM measurements.

The Z-potential value of the AgNPs colloidal suspension was also determined and resulted to be −39.0 ± 1.3 mV, suggesting that the colloidal suspension of the AgNPs is quite stable. A possible explanation of this negative Z-potential value of AgNPs is that the synthesis of NPs was carried out at pH 7, above the isoelectric point of quercetin, and it is possible that some quercetin molecules, negatively charged, remain absorbed onto the AgNPs surface. To confirm this hypothesis, it seems also that the quinone moieties of the QUC molecules are able to form chelates with good stability with the AgNPs, as described in the literature [35,36].

The features of the synthesized AgNPs are summarized in Table 1.

### 3.6. Antimicrobial Activity of AgNPs

The antimicrobial activity of AgNPs was investigated by determining the MIC value, defined as the lowest concentration of an antimicrobial that will inhibit the visible growth of a microorganism after overnight incubation [37]. The growth of three microorganisms typically involved in biodeterioration, namely, *Streptococcus* sp., a Gram-positive bacteria, *Escherichia coli*, a Gram-negative bacteria, and the fungus *Candida* sp., was followed by determining the absorbance at 600 nm (OD_600_) of the LB solution containing 100 μL of each bacteria and 100 μL of increasing concentrations of AgNPs with 8 and 20 nm diameter. The results are shown in Table 2. It is possible to note that AgNPs with a diameter of 8 nm are able to completely inhibit the microbial growth of all selected pathogens at a MIC of 1.0 μg/mL, while AgNPs with a larger diameter of about 20 nm are able to inhibit microbial growth of all selected pathogens at a higher MIC value of 2.5 μg/mL. The results obtained confirm that the antimicrobial properties of AgNPs are strongly affected by size, in particular the smaller the AgNPs diameter, the better the antimicrobial activity.

In particular, *Escherichia coli* seems to be the pathogen which is inhibited at the lowest concentration of AgNPs (0.5 μg/mL with 8 nm diameter AgNPs and 2.0 μg/mL with 20 nm diameter AgNPs), whereas the growth-inhibitory activity on *Streptococcus* sp. and on *Candida* sp. were definitely milder.

Table 3 shows a comparison between the MIC values obtained with AgNPs synthesized according to “green” processes utilizing different biological sources, such as plants, bacteria, fungi, yeast, and algae against different pathogens [38,39,40]. In particular, it is interesting to compare the results obtained against *E. coli* and *Candida* sp. with the AgNPs synthesized in the present study and those biosynthesized by *Rhodotula* sp. Strain ATL72 [7], which showed similar dimensions (8−20 nm diameter). The results are more or less similar, indicating a slightly higher bactericidal effect (lower MIC value of 0.5 μg/mL) with AgNPs synthesized using quercetin as a reducing agent (our work) with 8 nm diameter against *E. coli*, the other results being almost the same. As already remarked before, the AgNPs synthesized according to the green process developed in this study show very small average diameters compared to green AgNPs synthesized according to different processes and therefore they have a higher antimicrobial activity and lower MIC values against a wide range of pathogenic bacteria and fungi.

## 4. Conclusions

Silver nanoparticles were successfully synthesized according to a new green pathway using quercetin as a reducing agent in aqueous solution at room temperature. Compared to traditional methods, the new synthesis completely avoids toxic waste products and/or the use of harmful chemicals. Another important aspect is that the green process is cost-effective and time-saving as the synthesis occurs in only 20 min. The synthesized AuNPs were characterized in terms of size, morphology, and crystalline nature by utilizing different techniques. They resulted in spherical colloid particles with a crystalline structure and no remarkable aggregates or clusters and with different diameters depending on the metal precursor concentration.

Moreover, the antimicrobial effects of the green AgNPs were tested using three microbial strains usually present in biofilms associated with biodeterioration. The results support the findings that the synthesized AgNPs show potential antimicrobial properties, strongly influenced by the nanoparticle dimensions. In particular, the data showed that the MIC values were the lowest against all pathogens employed when the AgNPs diameter was the lowest, such as 8 nm.

Therefore, the proposed green synthesized AuNP could be promisingly utilized as an effective growth inhibitor of microbial colonization of artworks, such as bactericidal coatings on surfaces of different materials, with interesting applications in the field of conservation of cultural heritage.

## Figures and Tables

**Figure 1 nanomaterials-10-00909-f001:**
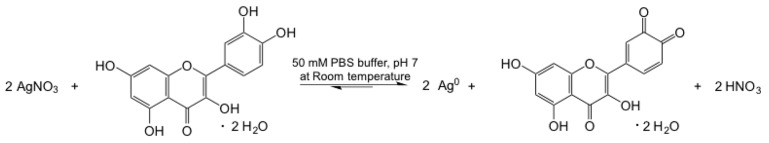
Synthesis of Ag nanoparticles through the reduction of AgNO_3_ by quercetin.

**Figure 2 nanomaterials-10-00909-f002:**
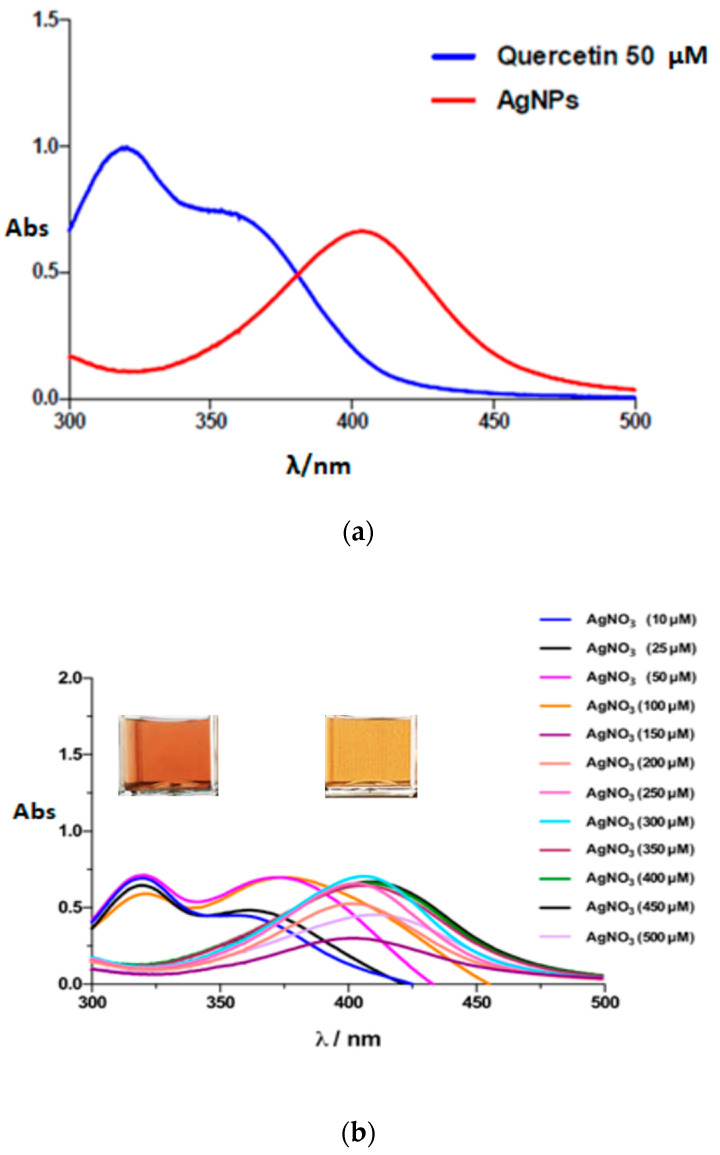
(**a**) UV-Vis spectra of a colloidal solution of QUC 50 μM in 50 mM PBS buffer pH (blue curve) and of AgNPs (diluted 1:4) (red curve); (**b**) UV-Vis spectra of a colloidal solution of AgNPs after 20 min (diluted 1:4) in 50 mM PBS at pH 7 obtained with different concentrations of AgNO_3_.

**Figure 3 nanomaterials-10-00909-f003:**
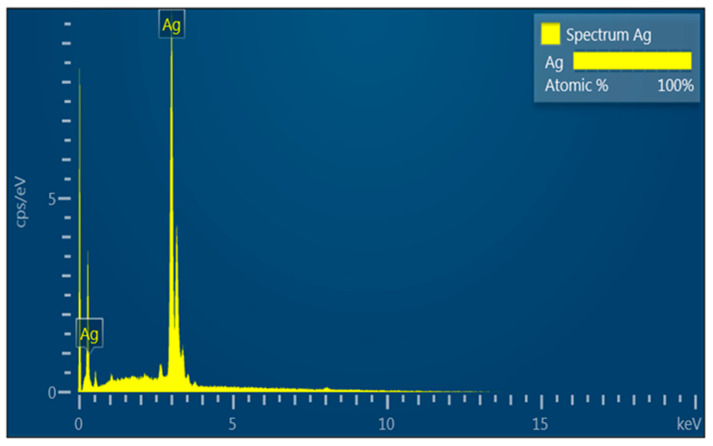
Energy dispersive spectra of a solution of AgNPs deposited on a copper grid.

**Figure 4 nanomaterials-10-00909-f004:**
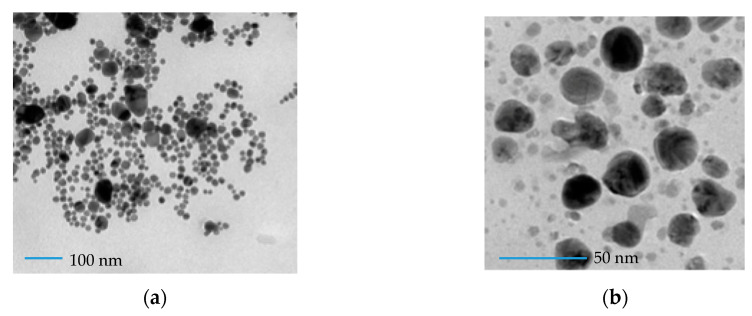
Transmission electron microscopy (TEM) image of a solution of AgNPs obtained with 350 μmoL metal precursor (**a**) and 500 μmoL metal precursor (**b**) deposited on a copper grid, and size distribution of a solution of AgNPs obtained with 350 μmoL metal precursor (**c**) and 500 μmoL metal precursor (**d**).

**Figure 5 nanomaterials-10-00909-f005:**
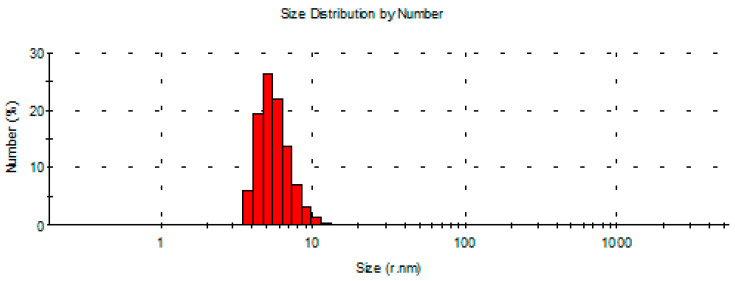
Size distribution of a solution of AgNPs (diluted 1:4) obtained with AgNO_3_ solution 350 μM in 50 mM in phosphate buffer (PBS) buffer of pH 7.

**Table 1 nanomaterials-10-00909-t001:** General features of the “green” AgNPs obtained with 350 μmoL metal precursor.

**TEM**	Diameter/nm	8.384 ± 0.287
**DLS**	Z-average/nm	10.01 ± 0.567
PDI	0.292 ± 0.049
Zeta Potential/mV	−39 ± 1.275

**Table 2 nanomaterials-10-00909-t002:** OD_600_ values and relative MIC of AgNPs with different diameters towards different pathogens.

AgNPs Diameter	8 nm	20 nm
Organism	*Streptococcus* sp.	*E. coli*	*Candida* sp.	*Streptococcus* sp.	*E. coli*	*Candida* sp.
MIC (μg/mL)	1.0	0.5	1.0	2.5	2.0	2.5
AgNPs Concentration (μg/mL)	OD_600_	OD_600_	OD_600_	OD_600_	OD_600_	OD_600_
0.0	0.65	0.52	0.60	0.65	0.52	0.60
0.25	0.40	0.08	0.32	0.58	0.45	0.55
0.5	0.22	0.00	0.15	0.50	0.30	0.45
1.0	0.00	0.00	0.00	0.31	0.16	0.28
1.5	-	-	-	0.25	0.10	0.22
2.0	-	-	-	0.10	0.00	0.15
2.5	-	-	-	0.00	0.00	0.00

**Table 3 nanomaterials-10-00909-t003:** Comparison of MIC values of “green” AgNPs from different biological sources against different pathogens.

Biological Sources	MIC (μg/mL)	Diameter (nm)	Pathogen	References
*Rhodotula* sp. Strain ATL72	1	8–20	*Bacillus* sp.	[7]
1	8–20	*E. coli*
1	8–20	*Candida* sp.
*Salvadora persica*	100	50	*E. coli*	[38]
400	50	*S. aureus*
*Streptomyces xianghaiensis* OF1 strain	16	64	*P. aeruginosa*	[39]
32	64	*C. albicans*
32	64	*M. furfur*
64	64	*B. subtilis*
64	64	*E. coli*
256	64	*S. aureus*
26	64	*Klesbiella pneumoniae*
Tea leaves	3.9	4	*Klesbiella pneumoniae*	[40]
3.9	4	*Salmnella Typhimurium*
3.9	4	*Salmonella Enteritidis*
7.8	4	*E. coli*
Quercetin	1	8	*Streptococcus* sp.	this work
2.5	20	*Streptococcus* sp.
0.5	8	*E. coli*
2	20	*E. coli*
1	8	*Candida* sp.
2.5	20	*Candida* sp.

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
