# Peer review of "Biocide Activity of Green Quercetin-Mediated Synthesized Silver Nanoparticles"

_nanomaterials, 2020, doi:10.3390/nano10050909_

Round 1

Reviewer 1 Report

A green synthesis of silver nanoparticles was synthesized by AgNO3 with quercetin. The synthetized AgNPs were characterized using UV-Vis spectroscopy, TEM, EDS and DLS techniques. Finally, the AgNPs were tested for biocide activity by studying their effects in the inhibition of bacterial growth. This kind of study is not new, the authors need to mention the novelty. In addition, there are some major aspects must be addressed.

  1. TEM image of a solution of AgNPs obtained with 500 uM is lack.
  2. The authors should mention what AgNO3 concentration was use in Fig 5. It will be better to exhibit both AgNPs’s DLS data were prepared by 300 uM and 500 uM of AgNO3.
  3. Why did the authors focus on using the AgNPs were prepared by 350 and 500 uM of AgNO3?

Reviewer 2 Report

The manuscript entitled “Biocide Activity of Green Silver Nanoparticles for Cultural Heritage Conservation” by Tasca and Antiochia, deals on the synthesis of silver NPs using quercetin. The paper is interesting but incomplete. In addition, I found some mistakes. In my opinion the paper should be improved in order to be published in “Nanomaterials

The authors did not implement an application for cultural heritage conservation, so the title must be changed.

Page 2, line 71: "It seems that the biomolecules present in plant extracts able to reduce the metal ions to form NPs are proteins, polyphenols, sugars and vitamins, such as L-ascorbic acid, caffeic acid and quercetin".

Another important molecule from plant extracts used to reduce silver is curcumin, please include some references.

Page 2, line 84-85: “Quercetin was solubilized in 1 M NaOH solution with a final concentration of 1 mM, then diluted in 50 PBS buffer at pH=7”; and line 87: “The synthesis was realized at room temperature in 50 PBS buffer”

It should be 50 mM PBS buffer.

Besides, the concentration of quercetin for synthesis is not clear; first, it was prepared a solution 1mM but later diluted in PBS, at which dilution?. In the caption of Figure 2, it is written that a solution QUC 50 mM in 50 mM buffer pH.

Page3, lines 112-113: "…and the Z-potential were determined by DLS experiments”

Z-potential and hydrodynamic diameter can be measured with the same instrument, in this case a Zetasizer Nano ZS90,but Z-potential is not a DLS experiment.

Page4. The author should include in Figure 2, images showing the color change of the samples as function of AgNO3 concentration.

Page 5.lines 163-165:” Moreover, the concentration of AgNPs colloidal solution has been calculated by using an extinction molar coefficient of 4.18 x 109 M-1 cm-1 at  λmax = 405 nm [25,26] and resulted to be 0.71 and  0.42 for 350 mM and 500 metal precursor, respectively.” 

I do not understand a concentration without units (0.71 and 0.42?). The same can be applied for the next statement :“500 metal precursor”.

Page 5, line 178: “the average diameter of the nanoparticles resulted to be 8.4 ± 0.3 and 20 ± 0.4” I belive the error given by the authors is very low for this type of synthesis and also observing TEM image in Figure 4, it its shows small but also some large particles. The authors should also illustrate this Figure 4 with the TEM image corresponding to the synthesis with 500 μmol of AgNO3.

Page 6, Particle size and zeta potential measurements. The authors should indicate to which sample corresponds the hydrodynamic diameter and the Number distribution plot in Figure 5. In addition, did they not measure Z-ave for all the samples? Authors should include a Table with the details of AgNPs synthesis, SPR (nm), size by TEM, hydrodynamic diameter by DLS and Z-potential.

Reviewer 3 Report

The manuscript by Tasca and Antiochia reports the synthesis of new silver-based nanoparticles, their characterization by UV-Vis spectroscopy, TEM, EDS and DLS and their antimicrobial effects towards Streptococcus sp., Escherichia coli and Candida sp.

The manuscript is clear and easy to follow. However, it is not clear to me, what is the advantage of newly synthesized nanoparticles compared to the already existing ones? There are many different silver-based nanoparticles synthesized by „green“ methods that exhibit excellent antimicrobial properties. Authors should analyze and present this literature. In addition, discussion and comparison of the antimicrobial effects of synthesized nanoparticles vs literature data is entirely missing.

Some technical remarks

p 87 „The synthesis was realized at room temperature in 50 PBS buffer at pH=7“ Do you mean 50 mM PBS?

P191: Please indicate polydispersity

Table 1: control (0 concentration of Ag nanoparticles) should be included.

Reviewer 4 Report

Review Report

The study “Biocide Activity of “Green” Silver Nanoparticles for Cultural Heritage Conservation” by Tasca et al. is very important because of its handling of novel green nanoparticle (NP) synthesis. “Green” NP synthesis is often intended to (1) be economic (2) reduce the release of toxic compounds to the environment, and (3) lower impacts on environmental microorganisms during application. The study aims to achieve “Green synthesis goals” and also prove that “Green” NPs have antibacterial properties via bacterial growth test.

The manuscript is well written and structured.

There are some points in the manuscript that the authors should consider in order to make the manuscript more convincing in stating that it is a new method or tool for producing “Green” Ag NPs.

Introduction

Line 39:

“so-called” – there is no reason to use quotation marks

Line 37 to 46: paragraph “It is well known …chemical reactivity”. 

Line 64:

“exposition” –> exposure?

Line 71 to 75: paragraph “It seems that the biomolecules present in plant extracts able to reduce the metal ions to form NPs are proteins, polyphenols, sugars and vitamins, such as L-ascorbic acid, caffeic acid and quercetin …”

The writers should add more literature into this paragraph about “Green” Ag NP synthesis, such as whether extracted biomolecules from plants, fruits or others were applied in the “Green” synthesis and how the experimental conditions were, such as temperature, duration, etc…This paragraph will help the writers compare their method to others and to prove later that theirs is a new method and what its advantages are.

Line 76-80: paragraph “The aims…cultural heritage”:

Is the study the first one to propose and apply a new method to produce Green Ag NPs? If yes, the writers should mention it here because in the Abstract line 17-18it is stated that  “In this paper “green” silver nanoparticles have been synthesized according to a new method and proposed as an alternative tool for control of microorganisms responsible for cultural…”.

 Experimental

Line 82: 2.1. Synthesis, purification and characterization of AgNPs

The authors should describe this more precisely. For example:

Line 83: quercetin - where did the authors obtain it?

Line 84: AgNO3 - where was it from?

Line 85: PBS - this is mentioned for the 1st time in the manuscript, and the authors should write its full name and components, if the authors made the buffer, otherwise stating where it was from.

Line 87: it could be made clearer whether the authors could provide temperature or a range of room temperature, e.g. 20 or 20-250C.

Line 90: Mili Q water, where was it taken?

Line 119: Streptococcus sp., Escherichia coli and Candida sp.? The authors should clarify their taxa and habitats: for example, if they are bacteria (Gram-negative or Gram-positive), fungi, algae, etc… in their habitant (soil, water…). Are they relevant to microorganisms in an environmental application (monument, cultural heritage)?

Line 120: LB broth, the full name of LB should be written first.

Results and discussion

Overall in this section, the authors should compare their “Green” Ag NPs method to other studies and ague that their method and Ag NPs obtained more advantages.

3.6. Antimicrobial activity of AgNPs

In Table 1: To Ag NP concentration should be added the unit (µg/mL)

1) Were microorganism cultures at the healthy phase, exponential or stationary, or in the dead phase after 48h?

2) What was the concentration of microorganisms for biocide activity study? Was the concentration relevant to the environmental application condition (monument or cultural heritage)?

3) The authors can perform serial fold dilution tests to get MIC more precisely. For example, E.coli could be inhibited at a lower concentration of 0.5 (ug/mL).  In this study, the MIC of E.coli was 0.5 because the test was started at 0.5.

4) The authors should discuss more their “Green” Ag NPs results compared to other studies. For example, how was MIC in this study compared to others with the same AgNPs sizes, with the same microorganism? It is important to have MIC at the lower concentration of AgPs so that materials will be applied less in a large-scale (in situ) application.

5) It is also important to have control samples of microorganism growth (without AgNPs).

Conclusion

Line 228, 237: Au NPs?

Overall, the paper could be considered for publication after the suggested revision has been made.

Reviewer 5 Report

Reviewer Comments to Author:

In manuscript entitled “Biocide Activity of “Green” Silver Nanoparticles for Cultural Heritage Conservation” Authors have described the AgNPs synthesis by application of quercetin, their physicochemical application and application as antimicrobial agents.

This paper should be revised according to the following comments.

  1. In introduction part Authors should additional information regarding the antohre green synthesis methods eg. by lactic acid bacteria e.g Appl Microbiol Biotechnol. 2017; 101(19): 7141–7153 or actinomycetes J Appl Microbiol. 2016 May;120(5):1250-63.
  2. Please add the information about sample preparation for DLS analysis. How author prevent the aggregation of AgNPs in this measurements? What about influence of solvents and pH on Your’s AgNPs?
  3. What about influence of day-light on synthesis of Ag nanoparticles through reduction of AgNO3 by quercetin. Did Author consider it?
  4. Please provide the XRD data of Your nano-system.
  5. Please provide in supplementary date photographs of inhibition zones obtained in antimicrobial tests.
  6. Why in the title is include application of this AgNPs in “Cultural Heritage Conservation”. Where are data proof it? The application of AgNPs against the portion of microbial isolates are not proof.
  7. What about the efficiency of synthesis ? How the AgNPs after synthesis was clean up for unreacted Ag ions?

In my opinion manuscript required the major revision before final acceptance. 

Round 2

Reviewer 1 Report

The revised manuscript did a little improvement. However, the Figures should redraw and make them embellishing.

Reviewer 2 Report

The authors reviewed the article according to the reviewers' suggestions. It can be accepted for publication in the present form.

Reviewer 3 Report

The manuscript is significantly improved after the revision. I recommend to publish it.

Reviewer 5 Report

The revision are satisfactory.